# The Physical Education Class Perceived by Schoolchildren from 6 to 8 Years Old Expressed through Drawings

**DOI:** 10.3390/children8080666

**Published:** 2021-07-30

**Authors:** Javier Cachón-Zagalaz, Déborah Sanabrias-Moreno, María Sánchez-Zafra, Amador Jesús Lara-Sánchez, María Luisa Zagalaz-Sánchez

**Affiliations:** Department of Didactics of Musical, Plastic and Corporal Expression, University of Jaén, 23071 Jaén, Spain; jcachon@ujaen.es (J.C.-Z.); mszafra@ujaen.es (M.S.-Z.); alara@ujaen.es (A.J.L.-S.); lzagalaz@ujaen.es (M.L.Z.-S.)

**Keywords:** physical education, children’s drawing, primary school, physical activity, analysis

## Abstract

Physical Education is one of the subjects that arouses the most interest in children. The aim of this study is to find out the opinion that primary school students have about the Physical Education class. Drawings from a sample of 62 students from an educational centre in the city of Jaén, aged between six and eight years old, were analysed. The results show that the larger size of the drawings corresponds to the aspects that are to be emphasised. This subject is carried out regularly in the sports pavilion of the centre, making frequent use of materials such as sticks, hoops or balls. Cheerful colours are used, reflecting their enthusiasm for the subject. The smiling facial expression represents the schoolchildren’s interest in the subject. The most popular games or sports are basketball and pichi, both of them collective.

## 1. Introduction

Drawing is not just a simple pastime or children’s game, but in fact involves the opening up of the child’s inner world, favouring fine motor skills and relationships with their environment. Throughout history, drawing has accompanied humankind, serving as a means of expression, the proof of which is cave art [1]. Drawing is the first means of communication to which children have access. This plastic or graphic representation allows children to express their emotions, interests and needs before being able to communicate them orally or in writing. Through drawing, they are able to to externalise the way they perceive and feel the context around them [2].

The practice of drawing in the most basic stages of education favours writing, reading and creativity, helps to develop self-confidence and to express the child’s feelings, among other aspects. Children’s drawing is, therefore, a rich means of communication as it offers significant information about themselves, others and the environment [3,4].

Likewise, the practice of physical activity (PA) at this age reinforces this learning and has an impact on the development of writing, creativity and other school subjects. Some authors [5] indicate that children draw their first lines between nine and ten months of age. Piaget [6] indicates that children begin to develop their ability to draw between the ages of two and seven, making their first monikers and drawings of elements at the age of three. As a consequence, fine motor skills begin to develop between the ages of one and four. 

Children’s drawings began to be valued at the end of the 19th century, considering this representation as artistic expression and education. At first, work focused on analysing children’s drawings in comparison with the creations of adults, accepting realism as the only possible explanation. Years later, in the 20th century, research expanded its expectations to include in the analysis of children’s drawings the expression of emotions and concepts of spatiality or creativity depending on the evolutionary process in which the artist was in [7]. Since then, there have been numerous investigations that have approached this subject from different perspectives.

In terms of the pedagogical approach, children’s drawing has gone through three stages: (1) traditional school, in which drawings were always made with a model present, they were copies and simple decoration; (2) free expression, where the painter Franz Cižek (1865–1946) grants minors the freedom of creation in their drawings, thus stimulating creativity [8]; (3) the stage of perceptive processes, seeking a more detailed analysis of the elements that make up children’s drawing. Between 1920 and 1970, scientific research on children’s drawing increased, with the most prominent in this period [9] focusing on the theory of the development of creative ability in children and establishing the stages of development of children’s drawing [10].

The stages that each child goes through in relation to drawing, according to Lowenfeld and Brittain [11] and Piaget [12] are:Scribbling stage: from 2 to 4 years of age. The child begins to make messy scribbles as a means of expression and to control movements and shapes as the infant grows older.Pre-schematic stage: 4 to 7 years. The child begins to be able to represent human forms. He distributes the elements he draws in space in a disordered way according to the importance he gives to each one of them.Schematic or logical realism stage: from 7 to 9 years of age. They draw more realistically, with more detail in the human figure, which makes them more recognisable to adults.Stage of visual realism: from 9 to 11 years of age. Children are looking for their drawings to be more realistic and begin to develop the ability to superimpose some elements on others.Pseudo-realistic stage: from 11 to 14 years of age. They already take spatial perspective into account. Their drawings become more complex.

Although Lowenfeld’s and Piaget’s stages are the most widely accepted in the educational field, Luquet [13] provides four other stages with different names, which he calls fortuitous, frustrated, intellectual realism (4–8 years) and visual realism from the age of 8 years onwards. Nevertheless, all three identify scribbling as a stage of drawing; the age they establish in the development of drawing is similar; they agree that the child tries to represent reality by drawing, and they think that big heads (human figures with limbs sticking out of their heads) are a common characteristic of the development of drawing.

In the educational field, drawing will allow the teacher to get closer to his pupils in order to detect significant information that he may not be able to obtain in any other way. Several studies have already used drawing as a tool to obtain and analyse information in relation to educational topics. Some authors [4] also analyse the way in which primary school pupils graphically represent children’s play in order to offer alternatives to the socially imposed models of physical culture in early childhood. An earlier study [14] investigated primary school pupils’ conception of school PE, showing particular attention to whether sport was included and the existence of possible sexist stereotypes in these classes. 

From 5 to 17 years of age, the World Health Organization (WHO) [15] recommends 60 min of moderate to vigorous intensity PA every day. Currently, the legal regulation in Andalusia (Spain) is 3 h of Physical Education (PE) per week in schools, one more than in other Autonomous Communities, so it is essential to make good use of them in order to achieve adherence to PA in schoolchildren [16].

The PE class is very important within the school curriculum, as it can influence the adherence that schoolchildren achieve and maintain towards PA [17]. It is therefore important to know the opinion that schoolchildren have about this subject. Pérez-Turpin & Suárez-Llorca [18] use children’s drawings to find out about different aspects of PE classes for students with motor disabilities. Moragón-Alcañiz & Martínez-Bello [4] also use drawing as a means of finding out the mastery of physical activity of pupils in the first cycle of Primary Education. Thanks to these studies, it can be affirmed that children’s drawing is a good way of measuring the opinion that children have about different concepts and, in this case, about the PE class.

The aim of the present study is to find out the opinion that primary schoolchildren have of the PE class.

## 2. Materials and Methods

The design is a cross-sectional, empirical–descriptive study using qualitative methodology through action research, the former to generate knowledge about the subject of PE and children’s views of their classes, and the latter to improve it if necessary [19]. The fundamental purpose of using action research in this study is to provide information to guide decision-making for programmes, processes and structural reforms [20], to improve pedagogical practices based on a thorough understanding of their specific characteristics.

The project complies with Spanish legislation and the international ethical standards established by the World Medical Association (WMA), which promulgated the Declaration of Helsinki in 1964 [21], although it has undergone subsequent revisions, the latest occurring in 2017.

The information transmitted to the families and teachers who are in direct contact with the study participants complies with the requirements of Spanish legislation in the field of research and personal data protection.

### 2.1. Participants

A total of 62 students (33 boys and 29 girls) from the first cycle of Primary Education at a public school in the city of Jaén took part. Their ages were between 6 and 8 years old.

### 2.2. Procedure

First, the researchers contacted the selected school to explain what the study consisted of and to get their approval. This is a public pre-school and primary school located in the city centre. When permission was obtained, the researchers went to the first and second year primary school classrooms to ask the children to draw a picture of what the PE class was like. The children were given complete freedom to draw whatever they wanted, as long as it was related to the requested theme.

### 2.3. Data Analysis

Initial categories were established for the quantification of the drawings and, after an initial review of the drawings, additional categories were defined for subsequent analysis. After this, we began to quantify each one of them, seeing if they appeared in each drawing and in what form. Initially, we thought of applying Cohen’s kappa coefficient, which is a statistical measure that adjusts the effect of chance on the proportion of agreement observed for qualitative elements (categorical variables), but the number of the sample allowed us to quantify the categories manually.

The categories that were decided upon are as follows: Size and proportion of human figures; Space represented; Colours used; Games or sports they draw; Clothes they identify with; Representation of movement; Main elements of the drawing; Number of figures present in the drawing; Facial expression.

## 3. Results

### 3.1. Size and Proportion of Human Figures

Children represent the elements they want to emphasise in their drawing at a larger size. As a general rule, they draw all human figures following the same proportion, without emphasising themselves with larger dimensions (Figure 1).

When they draw the teacher (N = 13), six students represent the teacher bigger, a detail that reflects the importance given to the teacher. Of these six, five are boys and the remaining drawing is of a girl. On the other hand, five draw the teacher at the same size as the rest of the people represented and only two girls draw him at a smaller scale (Table 1, Figure 2 and Figure 3).

There are four schoolchildren (one girl and three boys) who represent larger arms. In two of the drawings (Figure 4 and Figure 5) they are playing basketball and in the other two, the teacher is depicted solving conflicts between classmates. In these cases, it can be seen that they modify the size of the extremities depending on what they want to emphasise.

### 3.2. Represented Space

The space available at the school for PE classes is a gymnasium with a blue court and trellises surrounding the space. There are also two outdoor playgrounds, one of which is a sports court.

The distinction between indoors and outdoors at these ages can be seen in the colour of the court and in different elements represented, for example, the trellises in the school’s gymnasium. In both genders there are more representations of games inside the gymnasium than outside (20 boys and 19 girls). Only two boys depict the PE class on the outdoor track. Interestingly, no children depicted outdoor play with elements of the sky (sun or clouds). On the other hand, many of them specify that they do indoor PE by drawing the gym floor, adding the characteristic blue colour (Table 2).

Of the pupils (17 boys and 14 girls), 50% draw the floor of the pavilion occupying the whole sheet of paper, which shows that they graphically represent the space as they perceive it (very large), resembling their drawings to the reality that surrounds them. In the following images (Figure 6, Figure 7, Figure 8 and Figure 9), several representations of the interior of the pavilion can be seen, characterised by the use of the colour blue to represent the floor.

### 3.3. Colours Used

Children use bright colours for their drawings. None of them colour using black, dark blue or grey. Most of them choose warm colours to draw the physical-sports activities, which may mean that this subject brings them joy and fun. The most repeated colours are orange, red, green and blue.

### 3.4. Games or Sports They Represent

As can be seen in Table 3, the sport most represented by the schoolchildren in their drawings is basketball (N = 23), which is the most repeated by both girls (N = 10) and boys (N = 13). Basketball is followed by pichi (N = 10), a game similar to baseball, which is widely played in schools and is preferred by girls (N = 7 vs. N = 3). Following in descending order are Pacman (N = 8), dodgeball (N = 7), football (N = 6), hoop games (N = 3), crocodile (N = 2), butterfly catcher (N = 1), racket games (N = 1) and Minecraft (N = 1).

### 3.5. Clothes They Identify with

Four boys have designated the gender of the human figures in their drawing. They have done this by dressing the girls in skirts or dresses (N = 2) or by using different colours according to gender (N = 2). The girls who have painted their drawings with clothes (N = 10) do it by adorning them with dresses. In addition, it has been detected that they are more detailed and add elements such as bows, long hair or figures to their clothes (Figure 10 and Figure 11).

### 3.6. Representation of Movement

In some drawings (N = 9), the intentionality of drawing the movement of the sports or games represented can be appreciated. The illustration is done with additional strokes that indicate the direction or action. (Figure 3, Figure 12 and Figure 13).

Some drawings (N = 5) represent the people with respect to the original position they would occupy at the real moment of the game played. It can be seen in the following drawing how the persons occupying the goalkeeper’s position are drawn in a different position to the rest, even specifying that the goalkeeper on the right has thrown herself to the ground to stop the ball (Figure 14).

### 3.7. Main Drawing Elements

The following list represents the most frequently used elements in the drawings: baskets, balls, court, goals, hoops, backstops and cones. In most of the drawings, balls always appear, which helps to differentiate the game or sport they are representing. This may show that balls are one of the main objects they relate to the PE class. Cones and hoops also appear on many occasions, being, together with balls, the most abundant materials in schools.

In some cases, they are used as mere decoration and are not related to the activity being carried out, as in the case of trellises. No student identifies himself/herself in his/her drawing as playing or using espaliers, but 18 students represent them graphically.

### 3.8. Number of Figures Present in the Drawing

Of the students, 52 draw in the company of one or more people. Sometimes they also draw the teacher, although he/she is not usually part of the game. This may show that they consider the PE class as collective, not individualised, where they participate in the games and activities as a group and that the teacher is not integrated. Four girls and eight boys draw themselves, identifying themselves by indicating their name or ‘I’ (Figure 15 and Figure 16).

Only five drawings represent a single person. Figure 17 and Figure 18 show the representation of the space of a girl and a boy, respectively.

There are five schoolchildren who do not represent people in their drawings, generally focusing only on drawing their favourite game or sport (Figure 19).

### 3.9. Facial Expression

There are 50 drawings that have facial expressions drawn on people’s faces. Of these, in 46, happiness is evident. On the faces they draw big smiles, sometimes complemented by the representation of arms raised, which denotes enthusiasm (Figure 20, Figure 21 and Figure 22).

## 4. Discussion

The aim of this study is to find out the opinion that primary schoolchildren have about the PE class. This analysis has been carried out by interpreting the drawings made by the children and grouping them into different categories. In this section, we will discuss the main results obtained in the different categories.

### 4.1. Size and Proportion of the Human Figure

The children have represented in their drawings the largest proportion of the elements they wanted to highlight. This fact may be due to the fact that they do not yet know how to work adequately with proportionality between shapes. Melero-Merlo [7] states that the cause may be the subjectivity of the objects represented, understood as the importance or affectivity that they want to symbolise in these elements. Another of the reasons he suggests is that the child wants that particular object to be easily recognisable, which is why it increases its size. The same author uses the same theory to justify the fact that some children draw the teacher bigger. This is based on the affectivity they feel towards the teacher.

At the stage in which the children participating in the study are, they are already able to represent the human body, following a logical proportion between its parts. When some of these parts appear on a smaller or larger scale, it is for some reason. In the drawings of children playing basketball, the length of the arms indicates movement because of the importance of the technical gesture of shooting. The child feels “forced” to disproportionate this part in order to emphasise the function it performs, just as when they want to represent movement [7,22,23].

### 4.2. Space Represented

Most of the pupils drew elements of the gymnasium or the outdoor playground of the school, which allowed them to identify the area in which they represent the PE class. The colour of these elements was also very significant, since the gymnasium track is blue, so identification was very quick. Representing elements in the drawings with the colour they have in real life is a characteristic of the pre-schematic stage they are in [11].

Most of the pupils represented the PE class in the gymnasium and only two did so in the playground. This fact suggests that most of the sessions are held in indoor facilities, which has been corroborated by the school’s teaching staff. According to López-Moya & Estapé-Tous [24], it is not advisable to restrict activities to a single space, as this restriction may limit the teaching of the very diverse content of the area of PE.

50% of the pupils (17 boys and 14 girls) draw the floor of the pavilion occupying the whole sheet of paper, an aspect that shows that the children represent the space graphically at the same size as they visualise it, i.e., they see it as very large. At this age it is common for pupils to draw the environment and the objects found in it as close as possible to reality, specifying the colour or size, as they perceive it [2].

### 4.3. Colours Used

The use of colour in the drawing does not have an absolute meaning in its interpretation, but has to be evaluated as a whole, taking this aspect as a support for the analysis. However, the absence of colour can be understood, as those children with poor social skills or affective emptiness, who have less experience in using colour, draw less, and have inhibited emotional expression. Warm colours (red, orange or yellow) are associated with extroverted people, and cold colours (blue, green and purple) with introverted people. The preferred colours of pupils at this age are red, green, blue, yellow, violet, brown or black [25].

As can be seen, they are very mixed and do not show specific tendencies towards one or the other, although in the present analysis a large number of students choose cheerful colours for their drawings, an aspect which coincides with the authors mentioned and which may indicate that a large number of pupils in this class are extroverted and relate favourably with the rest of their classmates.

### 4.4. Games or Sports They Represent

The results obtained indicate that basketball is the sport most played by both boys and girls, probably because of the media coverage it receives, which coincides with [26] who point out that the games and sports that are best known and most played in schools are usually the ones that receive the most media coverage. However, it is curious that football, a highly mediatised sport, was only represented by six children. Likewise, there are not many gender differences in the choice of sporting activity, only football is represented by six boys and no girls, despite the fact that girls have taken up the sport and play it without problems and sometimes to a great extent.

At this point it can be thought that they are using a sign, which is a type of natural sign in which a relationship is established between the signifier and the signified; what is represented is present. For example, the drawing of a cloud is an indication of rain, and the drawing of a basket means that they are playing basketball; one thing is seen, and another is deduced. Meaning has a natural cause–effect relationship [27]. Perhaps emoticons would be part of the clue reinforcing the current culture, which is so entrenched in information and communication technologies.

### 4.5. Clothes with Which They Identify Themselves

In the analysis of the drawings, it has been detected that, in some cases, the female representation has been made through figures with long hair, dresses or skirts and accessories, such as handbags. These results agree with Pardo-Arquedo [28] who states that the representation of the female gender is usually reflected with these attributes and elements, which could mean that children have internalised the way of dressing according to the gender norms present in society.

Four boys specified the sex of the human figures in their drawing. They did this by dressing the girls in skirts (N = 2) or by using different colours according to gender (N = 2). Girls have put dresses on their drawings (N = 10). In addition, it has been detected that they are more detailed and add elements such as bows, long hair or drawings to their clothes. The fact that they do not differentiate between genders in their drawings may be due to the fact that sportswear is usually unisex.

### 4.6. Representation of Movement

Of the total number of drawings analysed, nine clearly represent the movement of figures or elements, such as balls. This characteristic is not typical of the evolutionary stage in which they are found. Quiroga [29] states that the representation of movement appears in the last stage of the evolution of children’s drawing, so it can be deduced that these nine schoolchildren have this ability more developed than their peers of the same age. According to Urraca-Martínez [27], from the age of five they begin to represent the movement of inanimate objects such as balls or stones, by including scribbles that simulate the trajectory of the object.

### 4.7. Main Elements of the Drawing

Most of the children represent the sports court of the school by painting it blue, which at first led the researchers to think that they were representing a swimming pool. After checking that they do not practise swimming during PE lessons and the photos taken during the visit to the school, it was observed that the blue colour was that of the gymnasium floor.

Several drawings show trellises, a classical gymnastic instrument found in most of the gymnasiums built in Spain in the second half of the 20th century and which, although we know that they are not commonly used, attract the attention of the schoolchildren who draw them [30,31]. Other portable apparatus frequently used in PE classes and which have been depicted by the children are balls, hoops and pikes. This is not the case with ropes, ribbons and balloons, which do not appear.

### 4.8. Number of Human Figures Present in the Drawing

It is significant that, despite the difficulties in realising the human figure that they have at this age, most children draw many people, which signifies the maintenance of the realism typical of the stage and the predominant socialisation in PE classes. The appearance of several figures implies the staging of their activity and the representation of movement [27]. Some authors [32,33] point out that the representation of movement is little studied in children’s drawing, although the need and interest in expressing an action graphically encourages the youngest children to modify the method they use to draw movement [22,23].

### 4.9. Facial Expression

In most of the drawings, smiles appear on the faces of schoolchildren. This, although at this age children are usually happy, means that they enjoy their PE lessons. Carcamo-Oyarzun [34] states that this fact is positive and contributes to their adherence to continued PA outside school. Generally speaking, and outside the different categories, it should be remembered that the ages of the participants are between 6 and 8 years old. They therefore comprise two different stages, pre-schematic and schematic. As can be seen from the figures in the manuscript, there are more developed and detailed drawings, and these usually correspond to older children who have already reached the schematic stage.

The study of the relationship that the children observe between PA and health has not been specified, because, being so young, it is not possible for them to attend to such a specific aspect of the subject; however, the research team understands that these studies lay the foundations for a later deeper knowledge on the part of the schoolchildren of these medical–social relationships. Nevertheless, in the previous (Section 4.9) reference is made to PA-Health when Carcamo-Oyarzun is cited. No previous article has been found that has already studied the vision that schoolchildren have of the PE class through drawings. Even so, it has been found that this is a good way of finding out about different aspects and opinions that children may have, and which they often do not know how to express. This is in line with the idea put forward by Mujica [35] when he states that drawing is a free and spontaneous means of expression to communicate what cannot be expressed verbally. Therefore, this type of work can help to improve different aspects of teaching using the vision of schoolchildren as a starting point. It would be interesting to continue studying along these lines, learning the opinion on other aspects and modifying some of the processes followed, such as, for example, having the children draw individually, separated from the rest of their classmates, so that they cannot copy elements from each other.

## 5. Conclusions

In response to the proposed aim, with the procedure used and the analysis of the drawings made by the schoolchildren, it is concluded that the children in the first cycle of Primary Education enjoy the PE class; they pay attention to details such as the characteristics of the facilities (colour of the gym floor, trellises, basketball hoops, sports equipment); the figure of the teacher who appears prominent and focal; they understand the socialisation that takes place in the session as they are surrounded by many classmates; they do not contemplate health issues because they are too young to understand that this is a quality of PA.

Specifically, and focusing on the sporting part, it is concluded that schoolchildren prefer collective games to individual ones, which allows us to understand that they have a group vision of this subject. The most commonly used sports elements in PE classes are balls, baskets, pikes, goals, backstops, hoops and cones, elements that can be found in any educational centre.

## Figures and Tables

**Figure 1 children-08-00666-f001:**
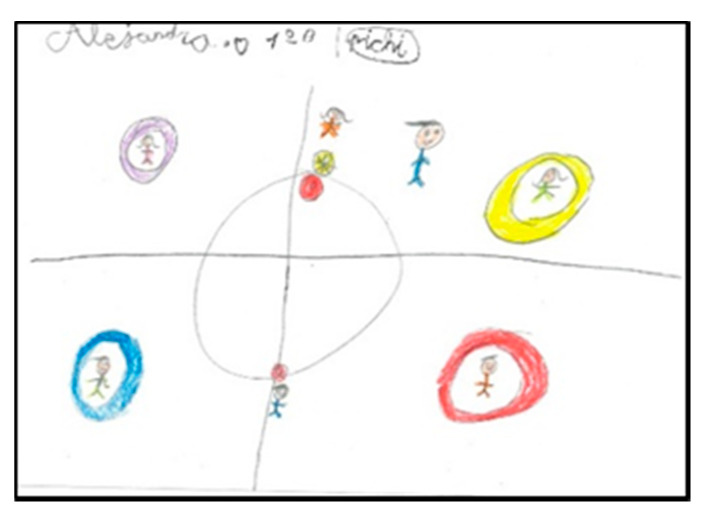
Children playing Pichi. Alejandro. 1°B.

**Figure 2 children-08-00666-f002:**
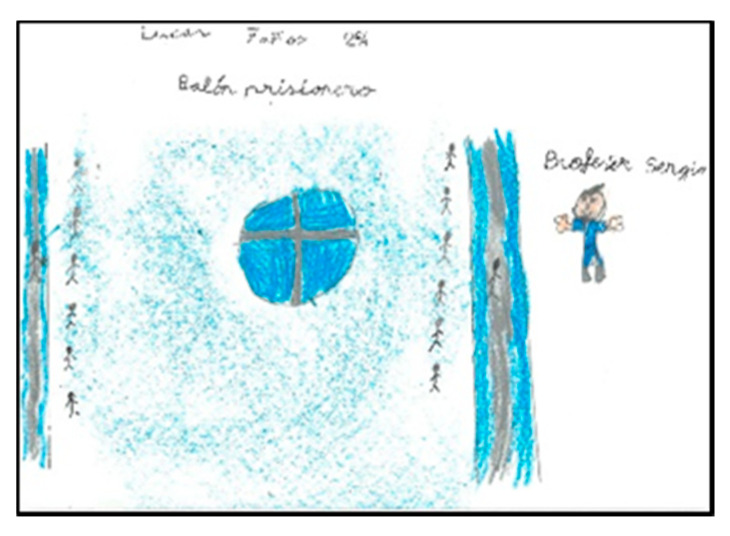
Larger teacher. Sergio. 2°A.

**Figure 3 children-08-00666-f003:**
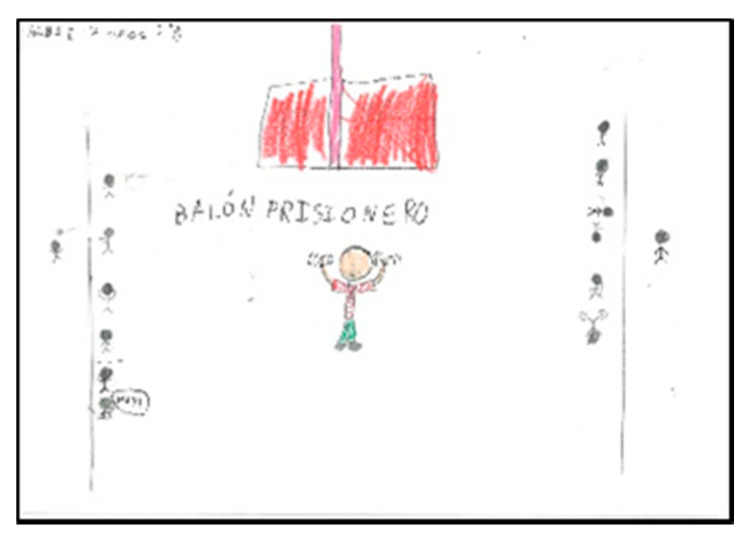
Teacher at a larger size. Alba. 2°A.

**Figure 4 children-08-00666-f004:**
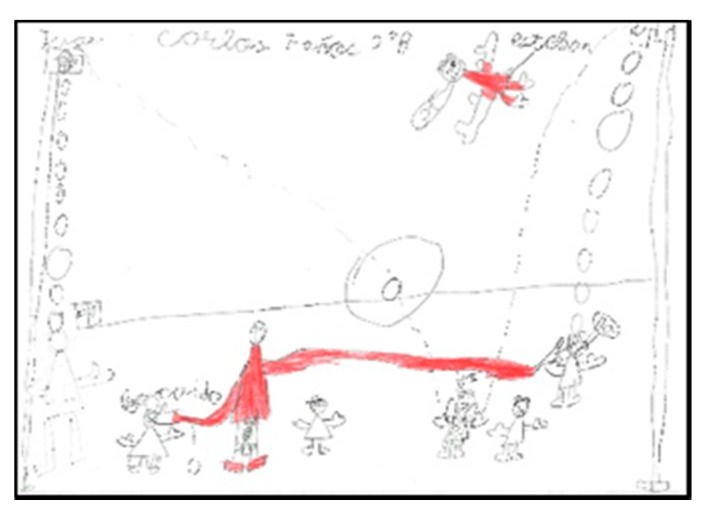
Human body proportion. Carlos. 2°B.

**Figure 5 children-08-00666-f005:**
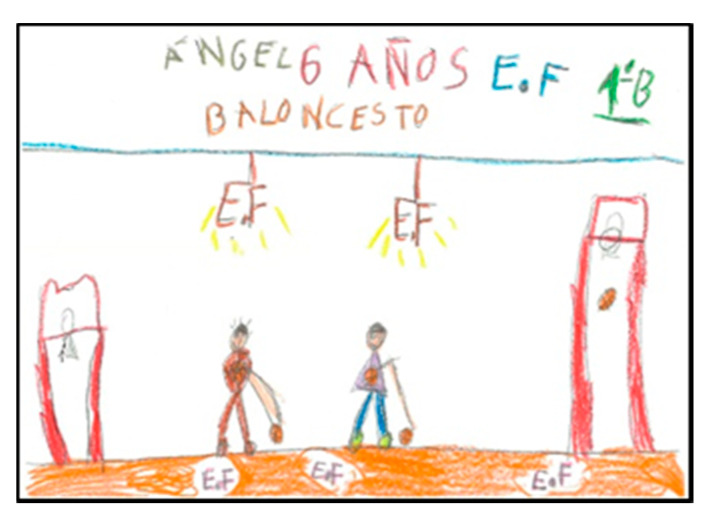
Human body proportion. Ángel. 1°B.

**Figure 6 children-08-00666-f006:**
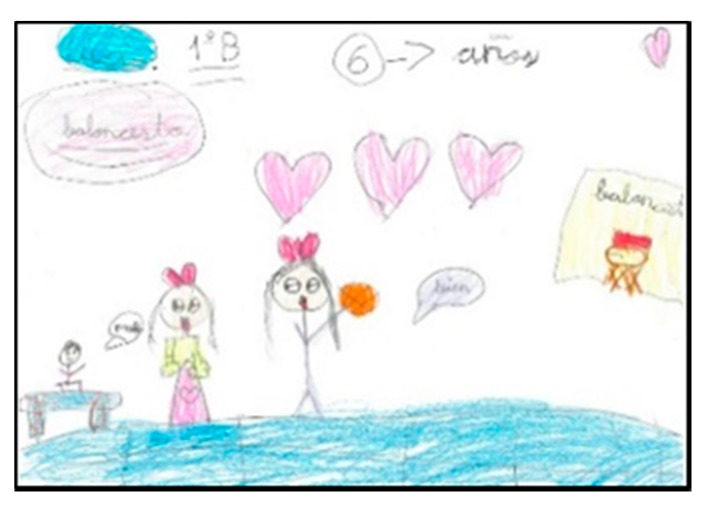
Represented space. Lucía. 1°B.

**Figure 7 children-08-00666-f007:**
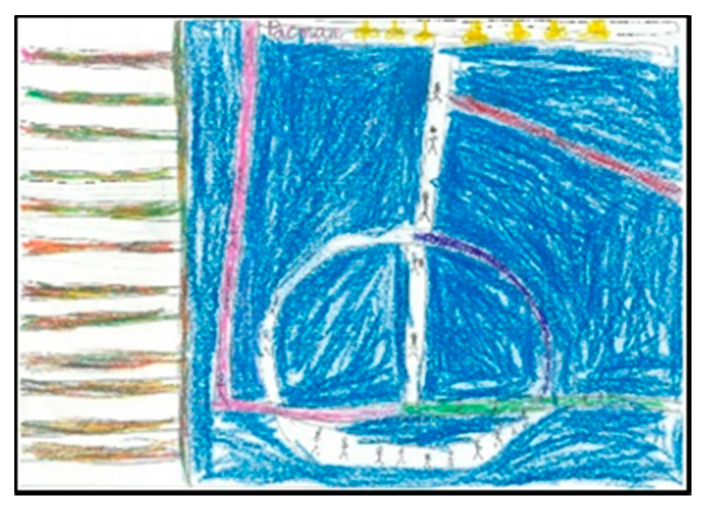
Represented space. Daniela. 2°B.

**Figure 8 children-08-00666-f008:**
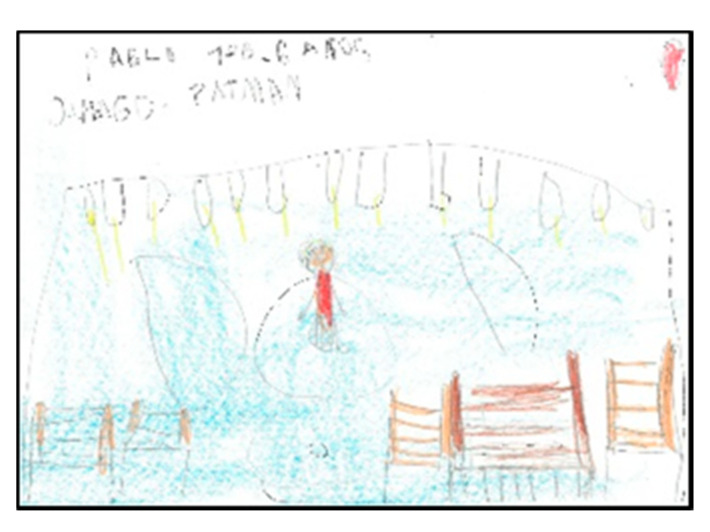
Represented space. Pablo. 1°B.

**Figure 9 children-08-00666-f009:**
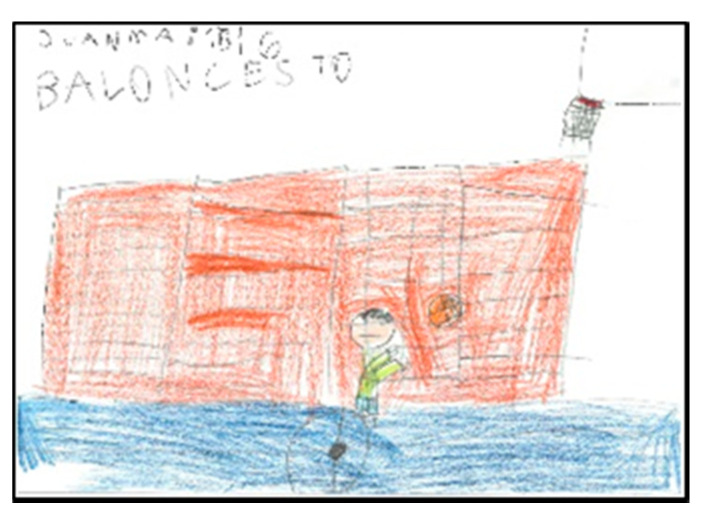
Represented space. Juanma. 1°B.

**Figure 10 children-08-00666-f010:**
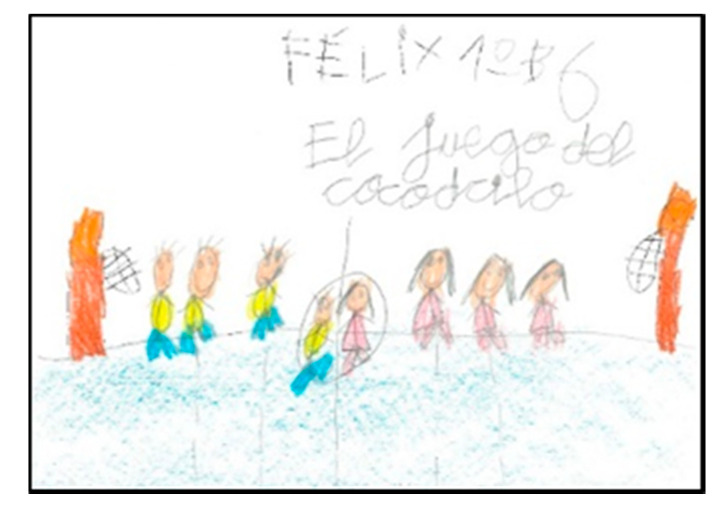
Clothes they identify with. Félix. 1°B.

**Figure 11 children-08-00666-f011:**
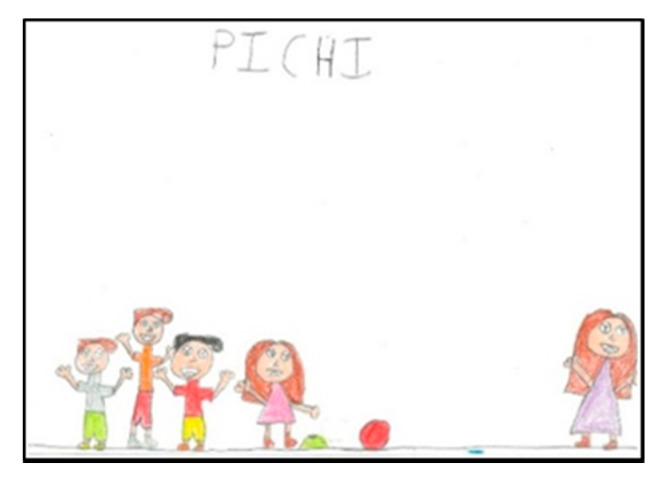
Clothes they identify with. Irene. 2°A.

**Figure 12 children-08-00666-f012:**
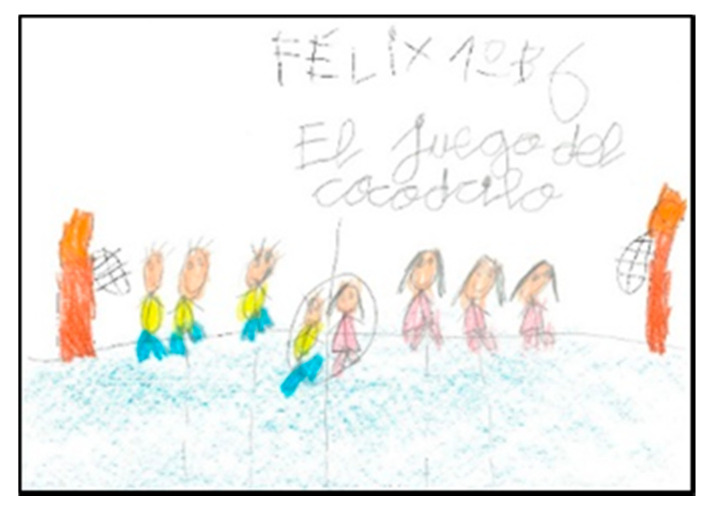
Representation of movement. Rodrigo. 2°B.

**Figure 13 children-08-00666-f013:**
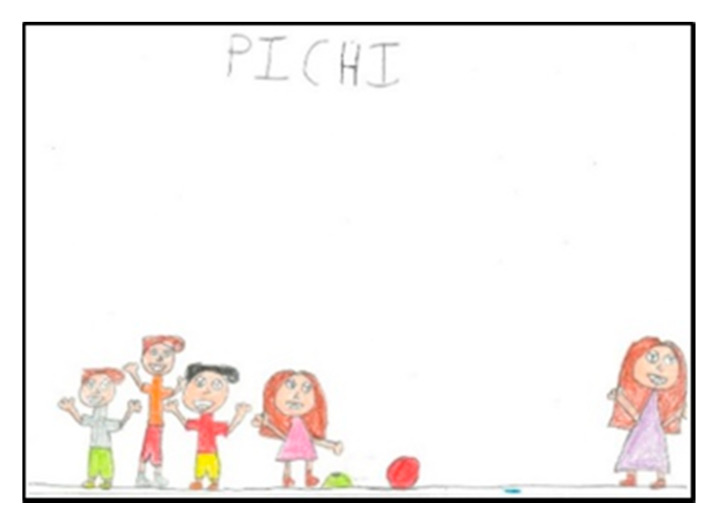
Representation of movement. Marcos. 1°B.

**Figure 14 children-08-00666-f014:**
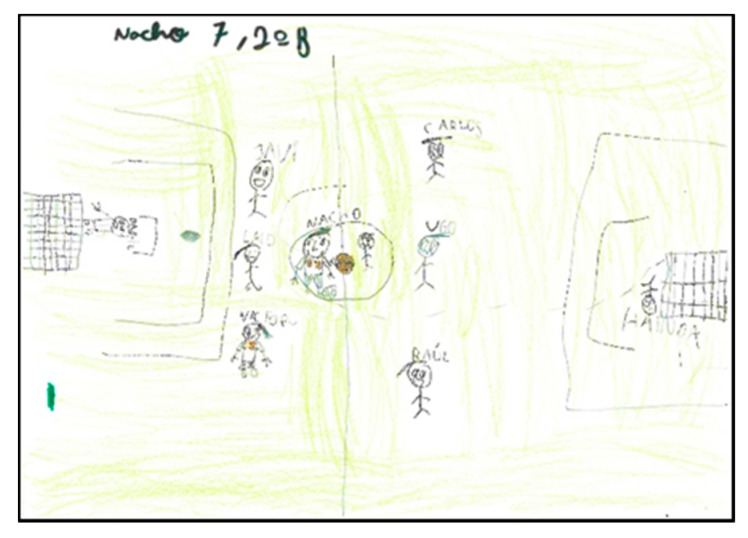
Representation of movement. Nacho. 2°B.

**Figure 15 children-08-00666-f015:**
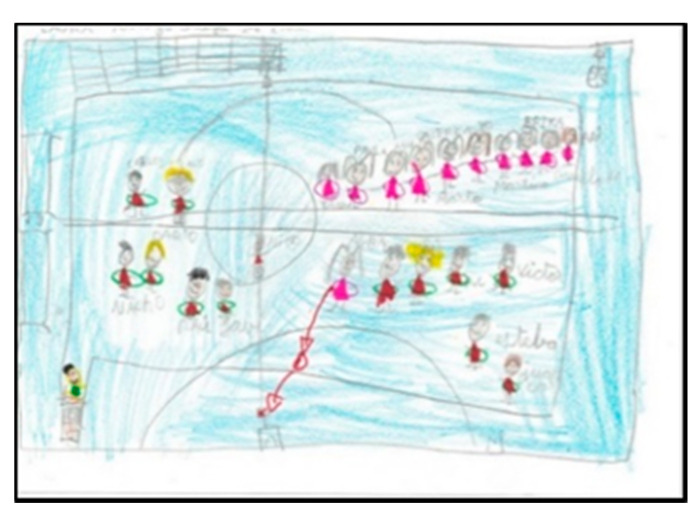
Number of figures present. Laura. 2°B.

**Figure 16 children-08-00666-f016:**
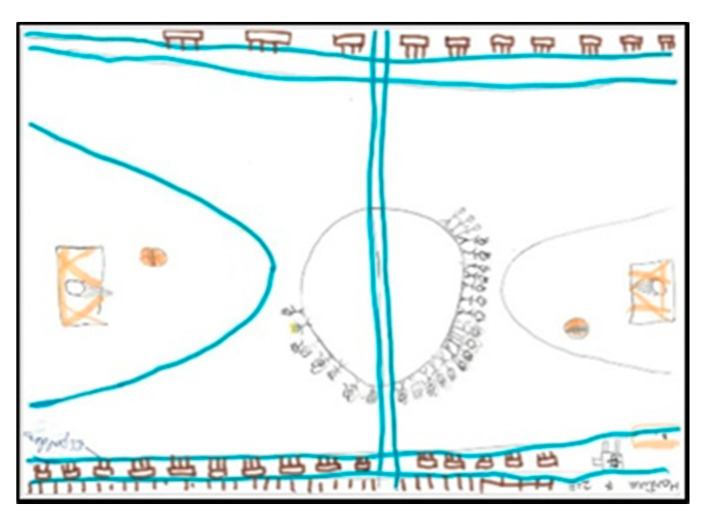
Number of figures present. Martina. 2°B.

**Figure 17 children-08-00666-f017:**
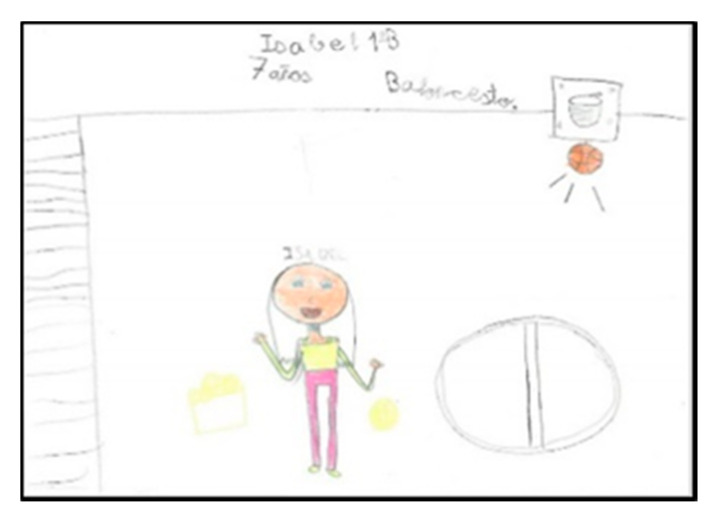
Number of figures present. Isabel. 1°B.

**Figure 18 children-08-00666-f018:**
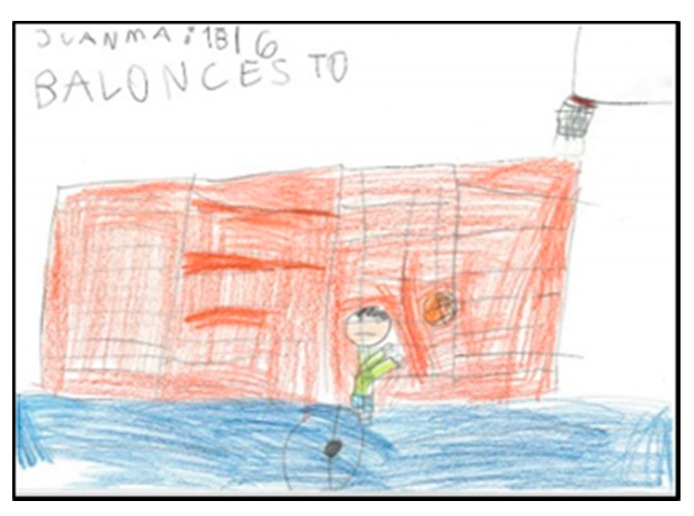
Number of figures present. Juanma. 1°B.

**Figure 19 children-08-00666-f019:**
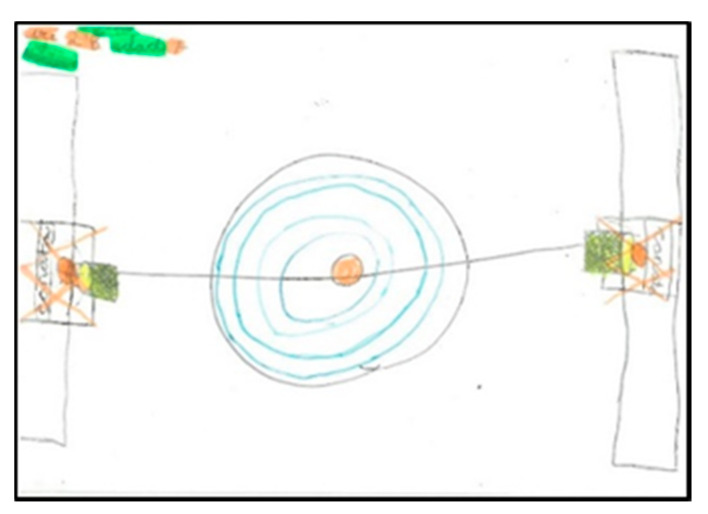
Number of figures present. Ari. 2°B.

**Figure 20 children-08-00666-f020:**
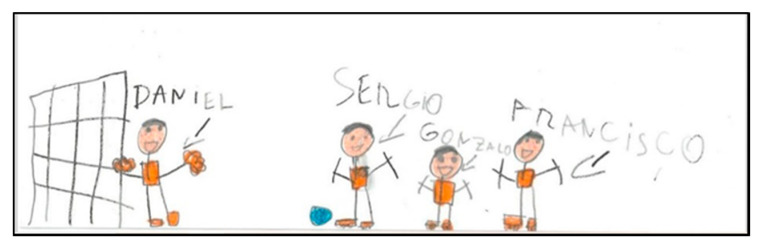
Facial expression. Sergio. 1°B.

**Figure 21 children-08-00666-f021:**
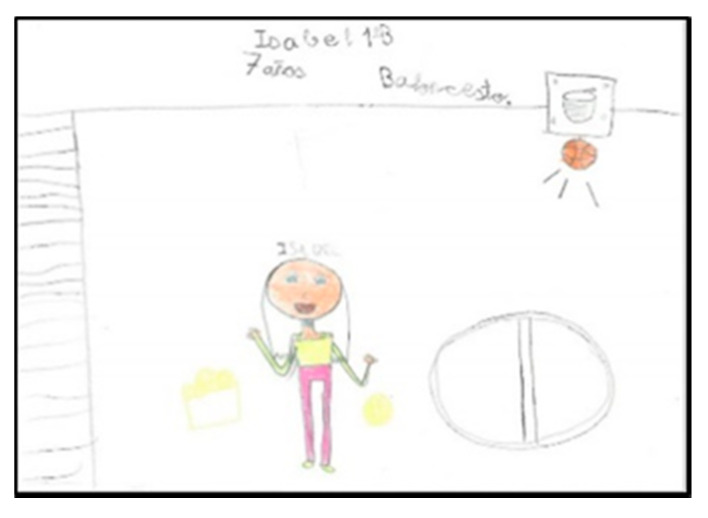
Facial expression. Daniel. 1°B.

**Figure 22 children-08-00666-f022:**
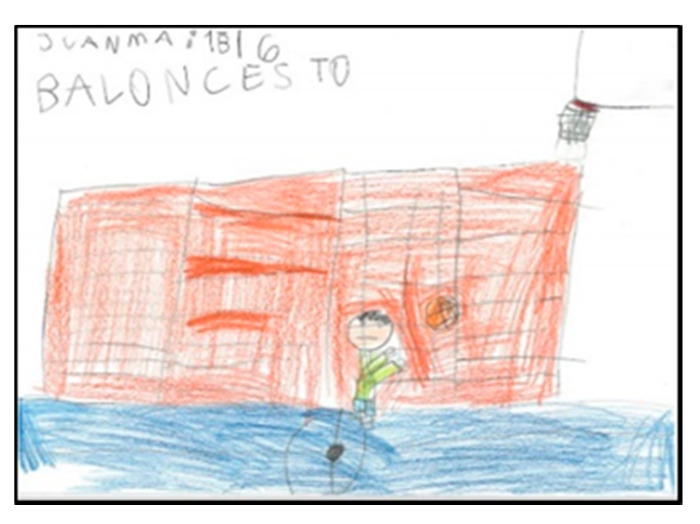
Facial expression. Esteban. 2°B.

**Table 1 children-08-00666-t001:** Representation of the teacher.

Size of the Teacher’s Figure in Relation to the Other Images	Boy	Girl
Largest teacher	5	1
Teacher of the same size	2	3
Smallest teacher	0	2

**Table 2 children-08-00666-t002:** Location of the space represented.

	Boy	Girl
Inside the pavilion	20	19
Outside the pavilion	2	0
Not distinguishable	11	10

**Table 3 children-08-00666-t003:** Games and sports represented by children.

	Boy	Girl	Total
Basketball	13	10	23
Football	6	0	6
Dodgeball	2	5	7
Pichi	3	7	10
Pacman or Pacman	6	2	8
Butterfly catcher	0	1	1
Tennis/Badminton	0	1	1
Hoop games	0	3	3
Crocodile	2	0	2
Minecraft	1	0	1

## Data Availability

The data are not publicly available due to the fact that they concern elaborations made by minors.

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
