# Peer review of "The Physical Education Class Perceived by Schoolchildren from 6 to 8 Years Old Expressed through Drawings"

_children, 2021, doi:10.3390/children8080666_

Round 1
Reviewer 1 Report
Thank you for the opportunity to review such a unique study. I found it most interesting that children’s drawings were utilised to understand their depiction and ‘possible’ perceptions of their Physical Education classes. Supporting children in their enjoyment of physical activity is an important aspect of their education.
The article reads well and each section was clear and well presented. The justification for the use of the drawing activity was well considered, however I did think that some further literature would actually strengthen this.
Some specific aspects for your consideration.
- The title – I did think that the word ‘art’ or ‘drawings’ could have been included in the title. This would aid future searches. The perceptions were gathered via drawings, so best to include this focus.
- The age of the children was 6-8 years and they were across two of the stages - pre-schematic and schematic. My suggestion is to develop a focus on this in the discussion.
- I actually like the sections in the discussion and these did explain the outcomes well, however these were reading more like results. Suggest that there does need to be more actual discussion. The answers to the following would enhance this further.
- How do you explain the results, particularly for the age group of your sample?
- Are there any similar studies that could be applied by way of comparison?
- What has been learned from this work that could continue to support positive involvement in this type of activity?
- What would further research look like? Is gathering children’s perceptions using drawing a useful activity that could be more utilised? Suggest this was a useful approach.
I agree with you the children were not focused on the health aspects and this is not required. However as educators, there is responsibility, therefore some emphasis on this would be also relevant. We do want children to develop sustained and lifelong enjoyment from physical activity.
Author Response
Please, see the attachment

Reviewer 2 Report
Review of the manuscript entitled: The Physical Education class as perceived by school children from 6 to 8 years of age. The manuscript submitted is appropriate to the subject matter and scientific rigor. The authors raised a very current issue at work, which is not only interesting from a scientific but also a practical point of view. Some remarks improving the quality of future research or suggesting an introduction to the submitted manuscript I send below in PDF file.
